# New Simulation Method for Dependency of Device Degradation on Bending Direction and Channel Length

**DOI:** 10.3390/ma14206167

**Published:** 2021-10-18

**Authors:** Yunyeong Choi, Jisun Park, Hyungsoon Shin

**Affiliations:** 1Department of Electronic and Electrical Engineering, Ewha Womans University, Seoul 03760, Korea; chidbsdud75@ewhain.net; 2Graduate Program in Smart Factory, Ewha Womans University, Seoul 03760, Korea

**Keywords:** flexible thin-film transistor (TFT), amorphous indium-gallium-zinc-oxide (a-IGZO), oxide TFT, bending stress, strain simulation, device simulation, channel length dependency

## Abstract

The dependency of device degradation on bending direction and channel length is analyzed in terms of bandgap states in amorphous indium-gallium-zinc-oxide (a-IGZO) films. The strain distribution in an a-IGZO film under perpendicular and parallel bending of a device with various channel lengths is investigated by conducting a three-dimensional mechanical simulation. Based on the obtained strain distribution, new device simulation structures are suggested in which the active layer is defined as consisting of multiple regions. The different arrangements of a highly strained region and density of states is proportional to the strain account for the measurement tendency. The analysis performed using the proposed structures reveals the causes underlying the effects of different bending directions and channel lengths, which cannot be explained using the existing simulation methods in which the active layer is defined as a single region.

## 1. Introduction

Flexible displays have attracted considerable attention as next-generation displays that are flexible, light, and not easily breakable [1]. As the active layer in flexible displays, many materials have been used, such as amorphous silicon [2,3], low-temperature polycrystalline silicon [4,5], semiconductor oxides [6,7], and organic materials [8,9]. Amorphous indium-gallium-zinc-oxide (a-IGZO) films offer the advantages of high mobility, small sub-threshold swing, low leakage current, and good uniformity owing to their amorphous phase, which makes them suitable for manufacturing large-area displays [10]. Moreover, a-IGZO thin-film transistors (TFTs) exhibit superior stability against bending stress than silicon-based TFTs [11].

The degradation of a-IGZO TFTs under bending stress has been studied extensively under various bending stress conditions, namely tensile or compressive stress [6,12,13]; bending direction perpendicular or parallel bending to the current flow [14,15]; bending radius [16,17,18]; and bending cycles [19,20]. Several research groups have extracted the density of states of the active layer by conducting computer-aided design simulations and verified that the number of donor-like trap states increases with repeated bending [15,21,22].

However, the existing research has not covered the effects of device geometry on the differences in terms of the magnitude of degradation and variation of density of states (DOS). Moreover, the general method of calculating strain assumes a simple one-dimensional structure and outputs a single strain value [23], irrespective of the bending direction and channel length. As can be inferred from the studies in the literature in which various bending radii have been used, a larger strain causes more severe device degradation [8,12]. Strain distribution is one of the important factors explaining this difference. Therefore, it is necessary to investigate the intensity and pattern of strain distribution in a device under each stress condition.

In this study, we conduct a mechanical simulation to obtain accurate strain distributions. Based on these distributions, a new device simulation method for flexible TFTs is suggested. The new method accounts for the dependency of device degradation on the bending direction and the channel length.

## 2. Mechanical Simulation Methods

As shown in Figure 1, a three-dimensional mechanical simulation structure was used to determine strain distribution. Multiple oxide and nitride buffer layers were placed alternately above the polyimide (PI) substrate. The bottom-gate electrode, gate insulator, and active layer were defined. An etch stopper was used to cover the a-IGZO, and the source/drain electrodes were placed on the left and right sides overlapping the active layer. Then, all these elements were passivated using an oxide film, except for the contact hole (Figure 1b). The material properties of each layer are summarized in Table 1.

The channel width was set to 50 µm, and various channel lengths of 10 µm, 30 µm, and 60 µm were used to derive different strain distributions depending on the channel length. The dimensions of the other elements were kept unchanged. The TFT was placed on a metal plate in the bending simulation. Two bending directions, perpendicular (Figure 1c) and parallel (Figure 1d), were considered, where the bending axis was either perpendicular or parallel to the current flow, respectively.

The plate was divided into several pieces in the same direction as the bending axis (Figure 2a,b) with different displacements assigned to each edge. As shown in Figure 2c, when the plate is divided into six pieces, the center of the plate is fixed, and the magnitude of displacement is calculated using the distance between the center, p0, and the edges p1, p2, and p3.

When the TFT is pulled along the length direction (*X*-axis) in the state of perpendicular bending, the displacement along the width direction (*Z*-axis) is set free. In addition, in parallel bending, the TFT is pulled along the width direction (*Z*-axis), the displacement along the length direction (*X*-axis) is set free. The bending radius (R) was set to 0.48 mm. In total, 580,539 nodes and 133,502 elements were used. The strain distribution on the bottom of the active layer was investigated by conducting a static simulation in the ANSYS software environment.

## 3. Results

### 3.1. Strain Distribution

The normal strain distribution at the bottom of active layer under bending is shown in Figure 3. The fact that the different strain level over the layer implies that the degradation of the a-IGZO film is nonuniform. Figure 4a,b show schematics of the active layer divided into nine regions based on the strain along each bending direction. The color of an area indicates its strain intensity, and the strain intensities increase in the order of yellow, orange, and red. The detailed values are investigated along the paths cutting the plane in the length or width direction, as indicated by the red lines (Figure 4c).

The overall strain distribution pattern differs depending on the bending direction. Under perpendicular bending, the strain is concentrated in the central part of the channel length (Figure 3a,c and Figure 5a), and there is no significant difference in strain along the width direction, as indicated by the flat curves in Figure 5b. In the case of parallel bending, the strain is the highest at the center of the channel, and it decreases close to the corners (Figure 3d–f). The strain curve is parabolic along the paths that cut the plane laterally (Figure 5c) or vertically (Figure 5d).

Bending stress leading to a strain of more than 2.2% over 100,000 repeated cycles was considered sufficient for cracking the active layer. In one a-IGZO TFT bending experiment described in the literature, it was mentioned that cracks occurred when a strain of approximately 2.17% was applied over 4000 cycles [24]. Moreover, the direction of crack propagation differed depending on the bending direction [20], as shown in Figure 4a,b. These results suggest that both strain and cracking affect the electrical properties of a-IGZO films, and the variation pattern of DOS can differ depending on crack orientation.

### 3.2. Device Simulation Structure

As shown above Figure 3, the magnitude of damage is nonuniform over the active layer. When the device is subjected to strain, the atomic arrangement of a-IGZO changes, and trap states are generated [12]. In particular, oxygen deficiencies serve as shallow donors of free electrons in the conduction band [25]. The increased donor-like states due to the ionized oxygen vacancies induce a negative shift in the threshold voltage [22,26]. When different levels of strain are applied to the a-IGZO layer, the corresponding changes in the trap states are different. Therefore, a uniform density of states with a single region is unrealistic in the case of a strained device.

Figure 6 shows the device simulation structures used to fit the transfer characteristics measured before and after 10,000 repeated bending cycles. In the case of the single-region structure (Figure 6a), the active layer is defined as a single region with a uniform density of states and is used in the device simulation before bending because of the lack of damage at that point. Two multi-region structures (Figure 6b,c), in which the active layer is defined as consisting of multiple regions with different densities of states in each region, are used for device simulation after bending.

First, the active layer subjected to perpendicular bending can be divided into the extensive, intensive, and extensive strain regions arranged in series (Figure 6b). The intensive region exhibits higher strain and has a greater number of donor-like states than the extensive regions. The transfer characteristics depending on the variation of trap states in each region are shown in Figure 7. The default curve is the simulation curve which is fit to the measurements of the device of channel length 10 µm after perpendicular bending. The words, ‘increased’ and ‘decreased’, in the legend means that the number of traps is increased or decreased by 5 × 10^16^ (cm^−3^) from the default concentration for acceptor-like and donor-like states, respectively, and the other parameters are the same as those in the default case. The variation of acceptor-like and donor-like states in the intensive region have little effect on transfer characteristic (Figure 7a,b) while the trap states in the extensive region control the threshold voltage (Figure 7c,d). These results indicate that the effect of the lower strain region is dominant in the perpendicular structure.

Second, under parallel bending, an a-IGZO film is divided into three regions (Figure 6c). According to the mechanical simulation results, it should be divided into at least nine location-dependent areas along the length and width direction (Figure 4b). However, because the low strain region determines the threshold voltage when a current flows through the high and low strain regions, as discussed in the perpendicular structure, regions near the source or drain have a dominant influence on the threshold voltage than the regions in the middle. Therefore, we focused on three areas in the first column near the source among the nine regions and simplified the parallel multi-region structure into three regions, namely the extensive and intensive regions placed in parallel (Figure 6c).

The Gaussian DOS parameters used for fitting the initial transfer characteristics before bending are 1 × 10^17^ (cm^−3^/eV), 3 × 10^16^ (cm^−3^/eV), 0.5 (eV), 0.25 (eV), 1.0 (eV), and 2.7 (eV) for the peak levels of density of states (NGA and NGD), their characteristic decay energies (WGA and WGD), and their peak energy distributions (EGA and EGD), respectively. The tail state parameters and band edge intercept densities, namely NTA ~5 × 10^19^ (cm^−3^/eV) and NTD 1 × 10^19^ (cm^−3^/eV), respectively, and the corresponding characteristic decay energies, namely WTA ~0.055 (eV) and WTD = 0.05 (eV), are used. The variation of DOS in the multi-region structure used to fit the measurements after the application of bending stress is discussed in the following section.

The two multi-region structures have different electrical properties owing to different arrangements of the multi-regions, as illustrated in Figure 8. The same proportions of multi-regions and the same density of states were used to compare the two multi-region structures. In the perpendicular multi-region structure, the extensive region had the dominant effect on the threshold voltage. By contrast, in the parallel multi-region structure, the intensive region had the dominant effect on the threshold voltage. This change in threshold voltage with the structure suggests that the bending direction is important, even when the same amount of strain is induced in the device. In the following section, we analyze the measurements using the proposed multi-region structures.

## 4. Discussion

The transfer characteristics of the devices with different channel lengths before and after 10,000 bending cycles are shown in Figure 9. The threshold voltage decreased after bending, and the amount of decrease under parallel bending was higher than that under perpendicular bending. This trend can be well calibrated using the proposed multi-region structures with density of states based on the strain distribution obtained in the mechanical simulation. Because the strain level is the highest in the central region of the device with the channel length of 10 µm under perpendicular bending, the highest peak level of donor-like Gaussian states (NGD), i.e., 1 × 10^18^, is applied in the intensive region of the perpendicular multi-region structure. Moreover, the smaller NGD of 7.3 × 10^17^ is applied in the intensive region of the parallel multi-region structure (Table 2).

The device simulation results indicate that the threshold voltage shift of the perpendicular multi-region structure is smaller, even though a larger NGD is used than in the case of the parallel multi-region structure. This result can be ascribed to the different arrangement of the multi-region structure. As current flows through the extensive and intensive regions arranged in series, both regions affect the current (Figure 10a), especially the extensive region. However, in the multi-region structure arranged in parallel, these regions act as independent current paths (Figure 10b), and the effect of the intensive region is not limited by the extensive region, which causes the large shift in threshold voltage.

In terms of channel length dependency, the shorter device exhibits a larger shift in threshold voltage. Under perpendicular bending, the strain near the source, which is the extensive region, increases as the channel length increases from 10 µm to 30 µm and, eventually, to 60 µm (Figure 5b). The proportional trap states are applied to the extensive region of the perpendicular multi-region structure, as summarized in Table 2, and the simulation results agree well with the measurements.

Moreover, in the case of parallel bending, the density of states was applied to the intensive region based on strain (Table 3). However, the device with L = 10 µm exhibited a larger degradation despite the smaller trap states than the devices with L = 30 and 60 µm (Figure 9b). This result can be ascribed to an increase in the activated density of states with narrower Gaussian donor-like states. To investigate the effect of the width of Gaussian donor-like states (WGD), the transfer characteristics were simulated using the same peak level of donor-like states and various WGDs (Figure 11a). The donor-like states applied in the a-IGZO film (solid line in Figure 11b) were not fully activated, and the peak level of the activated states (dashed line in Figure 11b) increased because the states were distributed over a narrower energy range. Thus, the device with L = 10 µm that used smaller and narrower donor-like Gaussian states exhibited a larger shift in threshold voltage.

In a comparison of the devices with L = 30 and 60 µm, the strain level at the channel center differed depending on the channel length. However, the difference decreased near the source (dashed line in Figure 5d). The fact that the devices with L = 30 and 60 µm exhibited similar shifts in threshold voltage can be ascribed to the same levels of strain near the source. Thus, the same amount of NGD was applied, and it calibrated the measurements well.

## 5. Conclusions

In this paper, we analyzed an a-IGZO TFT with various channel lengths of 10 µm, 30 µm, and 60 µm before and after it was subjected to bending stress. The strain distributions of the device under perpendicular and parallel bending were investigated by conducting a mechanical simulation. Considering the nonuniform strain distributions of the device, device simulation structures with multiple active regions were suggested for each bending direction. The different current flow mechanisms due to the arrangement of the intensively and extensively strained regions were analyzed. The proposed multi-region structures accounted for the measurement tendency whereby the device exhibited more severe degradation under parallel bending than that under perpendicular bending, even though the bending radius was the same in the two cases. Using trap states proportional to the strain, we explained the channel length dependency based on the good agreement between the simulation results and the measurements. Therefore, we expect that the proposed multi-region structures based on strain distribution can serve as a guideline in analyses of the effects of different bending conditions and device geometries.

## Figures and Tables

**Figure 1 materials-14-06167-f001:**
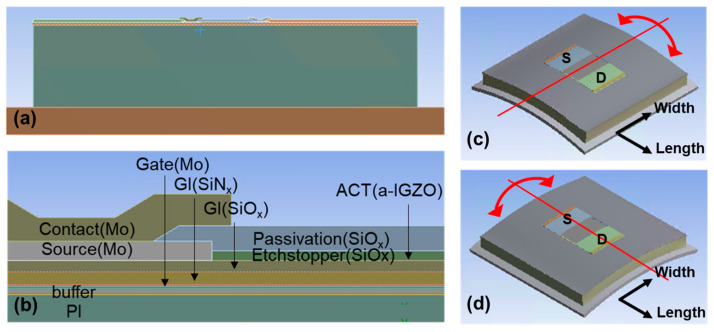
(**a**) Mechanical simulation structure, (**b**) detailed layers, (**c**) structure subjected to perpendicular bending with the bending axis vertical relative to the current flow, and (**d**) structure subjected to parallel bending with the bending axis parallel relative to the current flow.

**Figure 2 materials-14-06167-f002:**
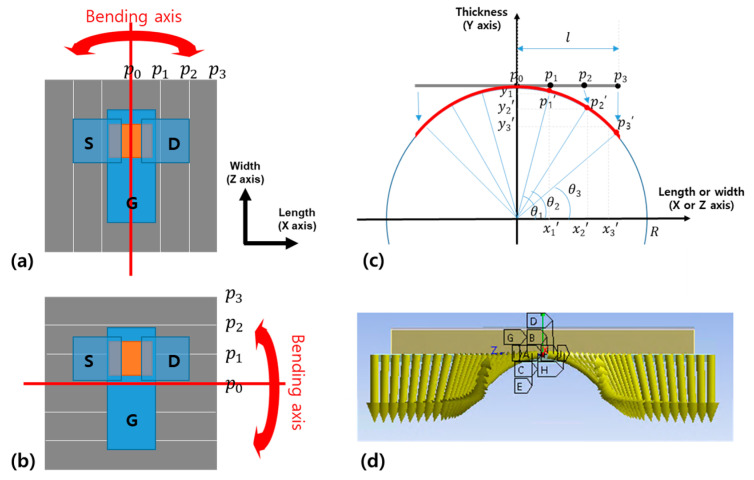
(**a**) Division of metal plate for applying perpendicular bending and (**b**) parallel bending to a-IGZO TFT. (**c**) Method for converting bending radius to displacement and (**d**) symmetrical displacement components used in ANSYS simulation.

**Figure 3 materials-14-06167-f003:**
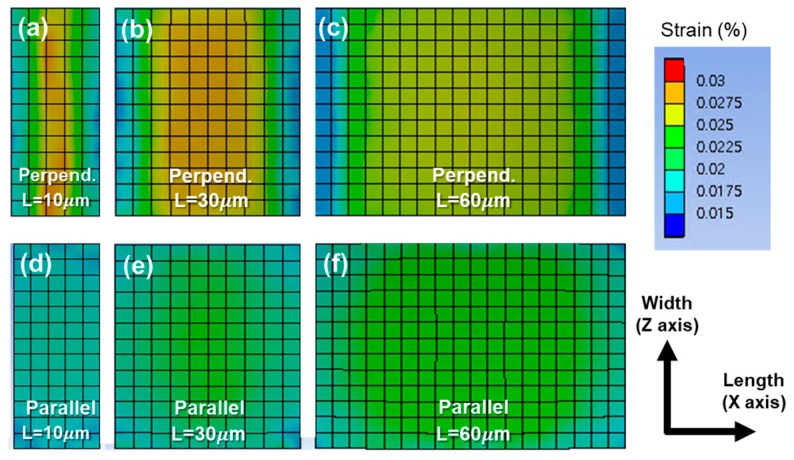
Strain distributions at the bottom of the active layer in a device with various channel lengths under perpendicular or parallel bending: (**a**,**d**) 10 µm; (**b**,**e**) 30 µm; and (**c**,**f**) 60 µm.

**Figure 4 materials-14-06167-f004:**
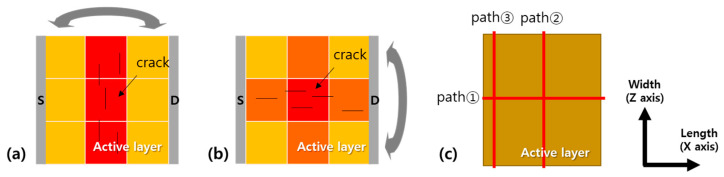
Division of active layer based on strain distribution under (**a**) perpendicular and (**b**) parallel bending. (**c**) Paths cutting the active layer laterally (path ①), vertically at the center of the channel length (path ②), or close to the source (path ③).

**Figure 5 materials-14-06167-f005:**
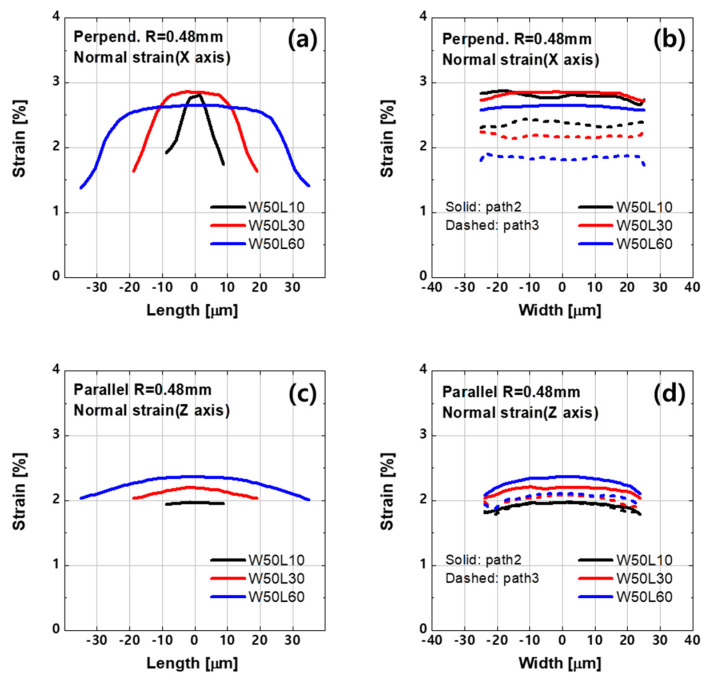
(**a**,**b**) Normal strain in the length direction (*X*-axis) under perpendicular bending. (**c**,**d**) Normal strain in the width direction (*Z*-axis) under parallel bending along the paths shown in Figure 4c.

**Figure 6 materials-14-06167-f006:**
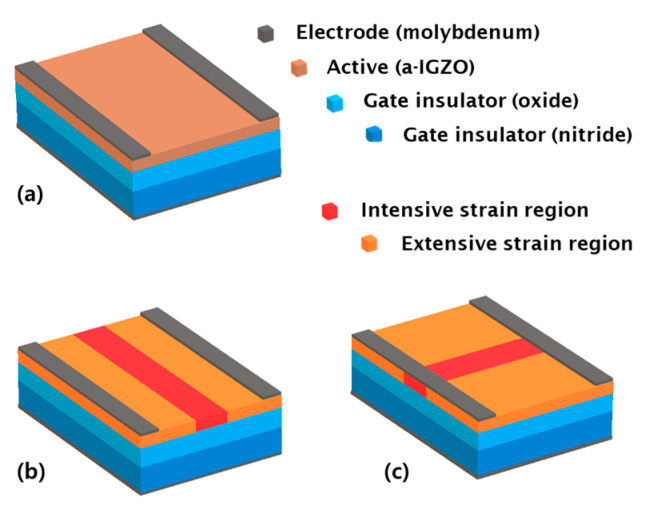
Schematic of device simulation structure: (**a**) Single-region structure with uniform DOS parameter for the active layer. (**b**,**c**) perpendicular and parallel multi-region structures consisting of intensive and extensive strain regions.

**Figure 7 materials-14-06167-f007:**
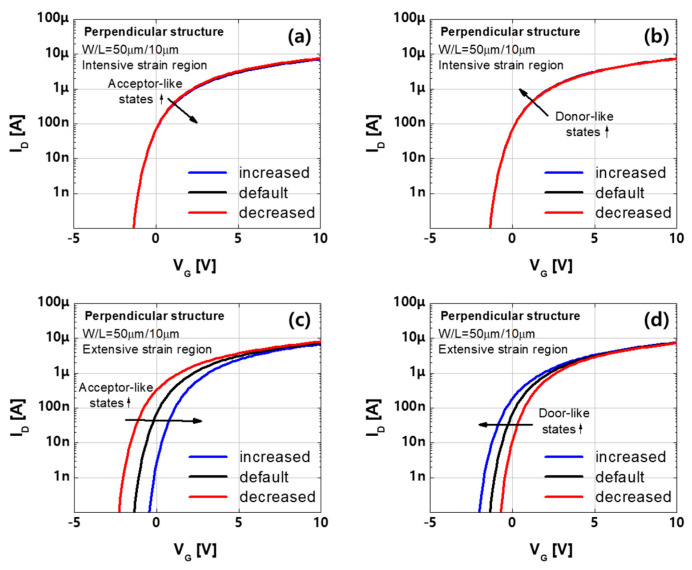
Effects of trap state variation in the (**a**,**b**) intensive and (**c**,**d**) extensive regions of the perpendicular multi–region structure.

**Figure 8 materials-14-06167-f008:**
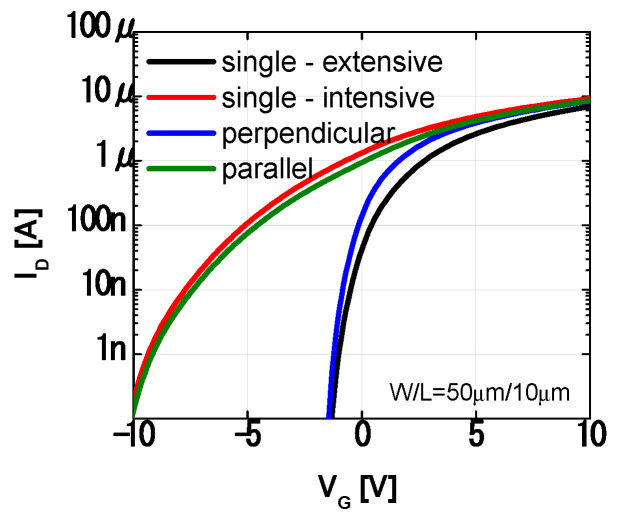
Simulated transfer characteristics of the multi–region structure, and two single–region structures with trap states in the extensive or intensive region.

**Figure 9 materials-14-06167-f009:**
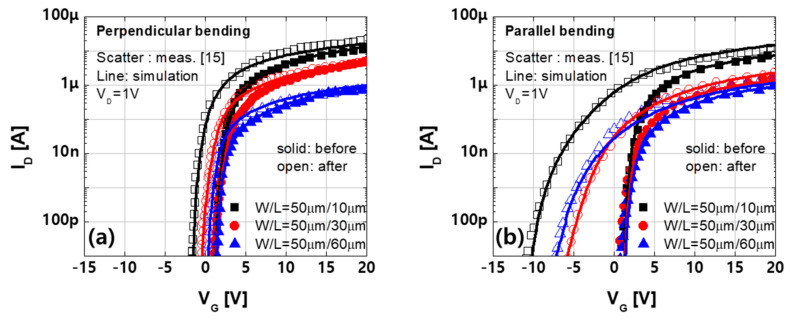
Measured (symbol) and simulated (line) transfer characteristics of the devices with various channel lengths of 10 µm, 30 µm, and 60 µm before and after (**a**) perpendicular bending and (**b**) parallel bending (V_D_ = 1 V).

**Figure 10 materials-14-06167-f010:**
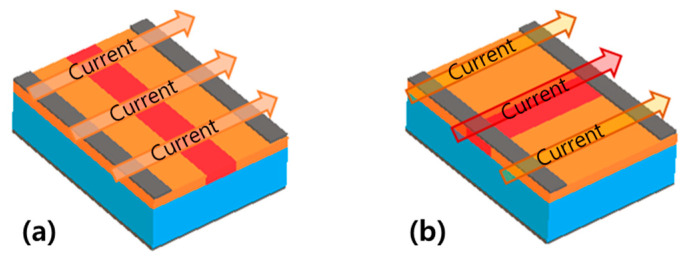
Different mechanisms of current flow depending on the bending direction: (**a**) perpendicular bending and (**b**) parallel bending.

**Figure 11 materials-14-06167-f011:**
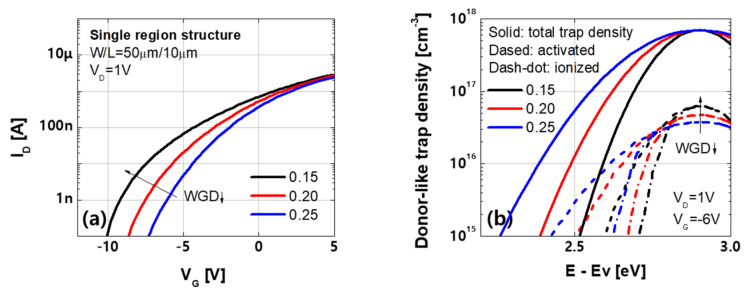
(**a**) Transfer characteristic with various widths of donor–like Gaussian states (WGD) (**b**) donor–like density of states as a function of energy at V_D_ = 1 V and V_G_ = −6 V.

**Table 1 materials-14-06167-t001:** Intrinsic parameters used in the mechanical simulation.

Layer	Material	Thickness [nm]	Young’s Modulus [GPa]	Poisson’s Ratio
Active	a-IGZO	20	130	0.36
Gate insulator1	SiOx	150	70	0.18
Gate insulator2	SiNx	100	250	0.25
Electrode	Molybdenum	60	315	0.30
Substrate	Polyimide	15000	2.5	0.40

**Table 2 materials-14-06167-t002:** Density of states of the parameters of the a-IGZO layer for fitting the measurements before and after perpendicular bending using the single- and multi-region structures, respectively.

Status	Structure	Region	Trap	Channel Length		
				10 µm	30 µm	60 µm
Before bending	Single-region	-	NGD ^1^	3.0 × 10^16^	3.0 × 10^16^	3.0 × 10^16^
			NGA ^2^	1.0 × 10^17^	1.0 × 10^17^	1.0 × 10^17^
After bending	Multi-region	Intensive	NGD	1.0 × 10^18^	1.0 × 10^18^	9.0 × 10^17^
			NGA	2.8 × 10^17^	2.8 × 10^17^	2.5 × 10^17^
		Extensive	NGD	4.0 × 10^17^	3.4 × 10^17^	2.0 × 10^17^
			NGA	2.2 × 10^17^	2.0 × 10^17^	1.6 × 10^17^

^1^ Peak level of donor-like Gaussian states. ^2^ Peak level of acceptor-like Gaussian states.

**Table 3 materials-14-06167-t003:** Density of states of the parameters of the a-IGZO layer for fitting the measurements before and after parallel bending using the single- and multi-region structures, respectively.

Status	Structure	Region	Trap	Channel Length		
				10 µm	30 µm	60 µm
Before bending	Single-region	-	NGD	3.0 × 10^16^	3.0 × 10^16^	3.0 × 10^16^
			NGA	1.0 × 10^17^	1.0 × 10^17^	1.0 × 10^17^
After bending	Multi-region	Intensive	NGD	7.3 × 10^17^	7.4 × 10^17^	7.4 × 10^17^
			NGA	1.0 × 10^17^	1.2 × 10^17^	1.2 × 10^17^
		Extensive	NGD	7.0 × 10^17^	7.2 × 10^17^	7.2 × 10^17^
			NGA	1.0 × 10^17^	1.0 × 10^17^	1.0 × 10^17^

## Data Availability

Data sharing is not applicable for this article.

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
