# Peer review of "New Simulation Method for Dependency of Device Degradation on Bending Direction and Channel Length"

_materials, 2021, doi:10.3390/ma14206167_

Round 1

Reviewer 1 Report

In this paper, the strain distributions of the device under perpendicular and parallel bending were investigated by conducting a mechanical simulation. However it has few grammar issues and should be improvement. This paper can be published in the Materials after addressing all grammar issues.

Author Response

We would like to thank you for reviewing our article entitled “New simulation method for the dependency of device degradation on bending direction and channel length”.

We have made several corrections and provided clarifications in the manuscript after addressing the reviewers’ comments. These changes are summarized in the attached Word file. 

Reviewer 2 Report

This paper explores how mechanical stress affect the electronic properties of flexible thin-film transistors (TFT) based on amorphous indium-gallium-zinc-oxide (a-IGZO) and shows it can be translated in terms of predictability of the behavior of such devices with time. As it is, the context is well explained and the results look convincing. The usefulness of such results is also established.

Unfortunately, the scientific justification of the results is nonexistent. The background of the numeric simulations is not explained, leaving a (hopefully false) feeling someone played at pushing button of a simulation without any physical knowledge of the whereabouts.

This paper deserves better. With the appropriate description of the simulation, it would become a sound article deserving publication.

Some minor editing is required: typo, punctuation, double apparition of a table, legends not attached to the figures...

See my remarks in the attached pdf

Author Response

(The authors gave the same response as above.)

Reviewer 3 Report

The authors reported a new simulation method for the dependency of device degradation on bending direction and channel length, and reveal the cause of the effect of different bending direction and channel length.On the whole, the work is relatively novel and reasonable. Before it can be accepted, some problems should be solved.

  1. Some literatures were out of date, authors should add some recent literatures.
  2. If possible, please give the formula of strain value given by simple one-dimensional structure and new device simulation method.
  3. In the introduction part, the authors mentioned that strain distribution is one of the important factors of the variation of DOS. It is better to give an explanation.
  4. In Figure 4(c), the location of source is not so clear.
  5. If possible, please explain the turn spots in Figure 5(d).
  6. Please check figure notes carefully, there are some mistakes. To be specific, the label of Table 2 is missing. In line 134, the semicolon need change into comma.

Author Response

(The authors gave the same response as above.)

Round 2

Reviewer 2 Report

Better version. Can be published as it is !